# Semicarbazide-sensitive amine oxidase inhibition ameliorates albuminuria and glomerulosclerosis but does not improve tubulointerstitial fibrosis in diabetic nephropathy

May YW Wong[1,2], Sonia Saad[2], Muh Geot Wong[2], Stefanie Stangenberg[2], Wolfgang Jarolimek[3], Heidi Schilter[3], Amgad Zaky[1], Anthony Gill[4,5], Carol Pollock[2]*

1 Department of Gastroenterology, Royal North Shore Hospital, St Leonards, Sydney, Australia, 2 Kolling Institute of Medical Research, Royal North Shore Hospital, University of Sydney, St Leonards, Sydney, New South Wales, Australia, 3 Pharmaxis Ltd, Frenchs Forest, Sydney, New South Wales, Australia, 4 Department of Anatomical Pathology, Royal North Shore Hospital, St Leonards, Sydney, New South Wales, Australia, 5 University of Sydney, Sydney, New South Wales, Australia

* carol.pollock@sydney.edu.au

**Data Availability Statement:** All relevant data are available at DOI: 10.6084/m9.figshare.12152625.

## Abstract

Semicarbazide-sensitive amine oxidase (SSAO) is an enzyme with a unique dual function in controlling inflammation as well as reactive oxygen species (ROS) generation. We have demonstrated benefit of SSAO inhibition in acute kidney fibrosis. However the function of SSAO in chronic kidney disease (CKD) and diabetic kidney disease (DKD) is yet to be determined. We aimed to assess the effectiveness of a SSAO inhibitor (SSAOi; PXS-4728A) as an antifibrotic agent using a diabetic model of CKD. Diabetic mice were treated with SSAOi for 24 weeks and outcomes compared with untreated diabetic mice and telmisartan treated animals as a standard of care comparator. Extracellular matrix markers, fibronectin and oxidative stress, were downregulated in diabetic mice treated with SSAOi compared with untreated diabetic mice. Expression of the pan-leukocyte marker CD45 was also supressed by SSAOi. SSAO inhibition in diabetic mice resulted in a significant reduction in glomerulosclerosis and associated albuminuria compared to untreated diabetic mice. However, the effect of SSAO inhibition was less obvious in the tubulointerstitial compartment than in the glomeruli. Therefore, SSAO may be a potential target for diabetic glomerulosclerosis.

## Introduction

Diabetic kidney disease (DKD) is the leading cause for chronic kidney disease (CKD) with an ever increasing rate. The management approach to DKD to date is restricted to the inhibition of the renin angiotensin aldosterone system (RAAS) [1], and more recently to blockade of sodium-glucose linked transporter-2[2]. However, this approach has limitations with ongoing risk of progression to end stage renal failure (ESRF). This manifests its impact on the

**Funding:** Pharmaxis (http://www.pharmaxis.com.au) supported the study in addition to a successful peer reviewed application to the Australian Research Council. WJ provided critical scientific review in the preparation of the manuscript but the content of the manuscript was at all times decided by the academic investigators. They approved the study design but were not directly involved. They were not directly involved with data collection or analysis. HS provided support with the SSAO activity assay. Both WJ and HS reviewed the manuscript prior to publication but were not involved in the decision to publish nor were they involved in the preparation of the manuscript.

**Competing interests:** MYW: None, MGW: None , SS: None WJ, AZ: None, AJG: None, CP, None SSaad: None, : Employee of Pharmaxis , HS: Employee of Pharmaxis. This does not alter our adherence to PLOS ONE policies on sharing data and materials.

healthcare system by increasing demands on renal replacement therapy such as dialysis and renal transplantation.

Clinical features which translate to a negative outcome include progressively increasing proteinuria and/or a reduction in glomerular filtration rate. Histopathologically, there is an expansion of glomerular basement membrane (GBM) and tubular basement membrane (TBM) as well as extracellular matrix (ECM) accumulation. Mesangial expansion ultimately results in glomerulosclerosis. As DKD progresses, there is the development of arteriolar hyalinosis, tubular atrophy and tubulointerstitial fibrosis. The ECM consists of collagen, fibronectin proteoglycans and glycoproteins. In physiological states, collagen IV is the most abundant collagen whilst collagen I and III are largely absent. During fibrogenesis, fibronectin is the initial ECM protein deposited which then activates integrins, attracts fibroblasts and co-localizes with collagen formation [3]. This results in the generation of more ECM proteins [4] and elastin [5]. Furthermore, the previously low levels of collagen I and III increases and is deposited both in the tubulointerstitium and in the glomeruli [6].

Semicarbazide sensitive amine oxidase (SSAO) is an enzyme predominantly located in the endothelium. It can also be found in large amounts in fat tissue, the liver, and gonads [7]. SSAO is active in the endothelial cells of vascularized tissues, including the kidney [7]. It is unique among other endothelial-expressed adhesins because it can also behave as an ectoenzyme. A soluble version of SSAO is found in plasma [8] and is known as vascular-adhesion protein-1 (VAP)-1 [9].

SSAO is involved in mediating inflammatory processes which can result in renal disease. The soluble end products of its enzymatic cleavage; hydrogen peroxide and reactive aldehydes lead to protein cross-linking and eventually oxidative stress [10]. SSAO also regulates the movement of leukocytes into areas of inflammation. Whilst initially this is a defensive and restorative process, when ongoing, can result in inflammatory cell accumulation. These two processes result in the development of renal fibrosis [11]. Inhibiting SSAO therefore appears to be a logical strategy to restrict inflammation and resultant downstream fibrosis in CKD. However, the direct role of SSAO inhibition in DKD is not well established [12].

We have previously studied the role of SSAO inhibitor as a treatment target in a model of unilateral ureteric obstruction (UUO) [13]. We found that SSAO inhibition is equivalent to standard RAAS inhibition in suppressing matrix gene expression, interstitial inflammation, oxidative stress, and total collagen accumulation. We demonstrated using an acute model of renal fibrosis that SSAO inhibition can effectively impair profibrotic and proinflammatory cytokine secretion, limit inflammatory cell accumulation, extracellular matrix expression, and oxidative stress. Although the UUO model is useful mechanistically as a model of acute fibrosis, the diabetic model is more relevant to human disease. The diabetic mouse model investigates both glomerular and tubular injury and may better simulate the functional and structural changes of CKD.

This study aimed to explore the role of an SSAO inhibitor in diabetic kidney disease, alone and in addition to standard therapy with RAAS blockade.

## Materials and methods

### Animal model

Approval was obtained from the Royal North Shore Hospital Animal Ethics Committee and abided to the Australian Code of Practice for the Care and Use of Animals for Scientific Purposes (ACEC#1309-003A). Mice were individually housed with free access to the rat and mouse premium breeder diet 23% chow (Gordons specialty feeds) and drinking water. Short

acting inhalational anaesthesia (2% isoflurane) was used for minor procedures as well as euthanasia at study completion.

Male endothelial nitric oxide synthase knockout (eNOS$^{-/-}$) mice on a C57BL/6 background were chosen as we have previously demonstrated their suitability to study DKD [14–19]. Mice were purchased from the Jackson laboratory (JR#002684 USA). Diabetes was induced using a low-dose streptozotocin (STZ) protocol with intraperitoneal injections of STZ (55 mg/kg diluted in 10mM citrate buffer, pH 4.5 given daily for 5 days, Sigma, MO, USA) at 7–8 weeks of age. Control mice received citrate buffer injections (pH 4.5). Blood glucose was obtained fortnightly using a glucometer (Accucheck Nano, Roche Diagnostics, North Ryde, Australia) following STZ through tail vein blood collection. Diabetes was defined as fasting blood glucose level (BGL) greater than 16 mmol/L. Long acting insulin (Insulin Glargine, Sanofi Aventis, Macquarie Park, Australia) was administered depending on the fasting BGLs of mice from 10 weeks of age. BGLs were checked monthly, or weekly if mice were receiving insulin or losing weight. One unit of Insulin Glargine was administered thrice weekly if BGL was greater than 28mmol/L and 2 units was given thrice weekly if BGLs exceeded 34mmol/L. The average amount of insulin administered across groups was recorded.

From week 12, the SSAO inhibitor (PXS-4728A;Pharmaxis Pty Ltd, Frenchs Forest, Australia) was administered using oral gavage (2mg/kg) via a disposable plastic gavage tube (Instech Laboratories, Pennsylvania, USA) with the vehicle phosphate buffered solution (PBS). (usually 4 weeks after STZ induction). Telmisartan (Boehringer-Ingelheim, Germany) was mixed with drinking water (0.008mg/mL, pH 7.4). The treatment groups were (1) Control (Ctrl), (2) Control receiving SSAOi (Ctrl+ SSAOi), (3) Diabetic (DM), (4) Diabetic receiving SSAOi (DM + SSAOi), (5) Diabetic receiving Telmisartan (DM + Telmisartan), (6) Diabetic receiving Telmisartan and SSAOi (DM + Telmisartan + SSAOi). There were 6 to 8 animals per group. On average, diabetic mice drank similar amounts independent of the treatment they received. Hence dosing was similar in groups 5 and 6. The pH of the drinking water was adjusted to 7.4 using 1% hydrochloric acid. Mice were culled 24 weeks after receiving STZ.

**Measurement of physiological parameters.** Body weight was recorded at baseline and animal sacrifice. Metabolic cages were used to collect 24 hour urine prior to sacrifice. Urine was collected terminally using bladder puncture. Spot urine creatinine was measured using a picric acid method (Creatinine Companion, Exocell Inc., Philadelphia, PA, USA) and spot urine albumin was measured using ELISA (Albuwel, Exocell Inc., USA). From these values, an albumin to creatinine ratio was obtained [15]. Any remaining urine was centrifuged and stored at -80˚C. Blood was obtained at time of sacrifice using cardiac puncture. Plasma aliquots underwent high speed centrifugation for 10 minutes at 4˚C. HbA1c was measured from a pre-terminal blood collection using a DCA Vantage analyser (Siemens Healthcare, Bayswater, VIC, Australia). Remaining plasma was snap frozen using liquid nitrogen and stored at -80˚C.

## Renal histology and morphometric analysis

After harvest the kidney was perfused with PBS. One half was placed in 4% neutral buffered formalin overnight followed by 70% ethanol. Wedges were then embedded in paraffin and cut onto glass slides (3μm). Masson's trichrome and Periodic Acid Schiff (PAS) staining was performed. The kidney cortex was examined under a light microscope (Olympus photomicroscope linked to a DFC 480 digital camera). Two blinded independent assessors; AJG and MGW reviewed histological sections and scored tubulointerstitial fibrosis and glomerulosclerosis. Tubulointerstitial fibrosis was characterised as tubular atrophy or dilatation, presence of mononuclear inflammatory cells, widening of interstitial spaces with deposition of ECM,

interstitial cell proliferation and wrinkling or thickened tubular basement membrane. Masson's trichrome staining was used to determine tubulointerstitial fibrosis. The tubulointerstitial damage index (TDI) was calculated using a scale of 0 to 4 as we have previously described [15,20,21]: 0—normal; 1—involvement of < 10% of the cortex; 2—involvement of 10–25% of the cortex; 3—involvement of 25–75% of the cortex; and 4—extensive damage involving > 75% of the cortex. Ten non-overlapping fields were studied. At least 5 glomeruli were included in each field. PAS staining was used to grade glomerulosclerosis: 0—normal, 1 - < 25% involvement, 2 < 50% involvement, 3 - < 75%, and 4 - > 75% sclerosis). A semi-quantitative glomerulosclerosis index (GSI) score was also calculated from averaging scores from all counted glomeruli in one section. We used interclass correlation coefficient (ICC) to compare the semi-quantitative scores obtained by assessors.

## Reverse Transcription Polymerase Chain Reaction (RT-PCR)

The other half of the kidney was snap frozen with liquid nitrogen. Total RNA was extracted from the kidney cortices (Qiagen, RNeasy Mini Kit). Pre-designed primers for fibronectin, collagen IV, transforming growth factor- β1 (TGF-β1), monocyte chemoattractant protein-1 (MCP-1), and actin were used (Table 1). A mastermix (Stratagene, La Jolla, CA, USA) containing 0.25μL of 10μM forward primer, 0.25μL of 10μM reverse primer, 5μL SensiMix SYBR Hi-ROX (Bioline, Eveleigh, NSW, Australia) was added to cDNA template along with DNA/nuclease free water to make a final reaction volume of 10μL. Reactions were performed in triplicate using ABI Prism 7,900 HT Sequence Detection System (Applied Biosystems, Mulgrave, Vic, AU). All data are presented as fold-change compared with control after normalization to the housekeeping gene β-actin.

## Immunohistochemistry

**Paraffin embedded sections.**   Paraffin embedded sections were dewaxed followed by antigen retrieval and blocking (Dako, Glostrup, Denmark). Primary antibodies against fibronectin (dilution 1:1000) (Sigma-Aldrich, Dublin, Ireland), collagen IV (dilution 1:1000) (Abcam, Cambridge, UK), pan-leukocyte common antigen CD45 (dilution 1:100) (Merck Millipore, Darmstadt, Germany) and nitrotyrosine (1:400 dilution, Upstate Biotechnology, Temecula, CA) were applied at 4°C overnight followed by incubation with secondary anti-rabbit Envision-system (streptavidin-biotin (LSAB+) detection system; Dako Glostrup, Denmark). Counterstaining was performed with Mayer's Haematoxylin (Fronine, Taren Point, NSW, AU) followed by Scott's solution (Fronine, Taren Point, NSW, Australia). Control sections were prepared with an irrelevant isotype matched IgG.

The slides were examined using a light microscope as above where ten to twelve consecutive non-overlapping fields from each section of renal cortex and glomeruli were photographed.

**Table 1.  Primers sequence for mice fibronectin, TGF- β1, collagen IV, MCP-1 and actin genes.**

| Gene Name | Forward Primer (5'-3') | Reverse Primer (5'-3') |
|---|---|---|
| Fibronectin | CACGGAGGCCACCATTACT | CTTCAGGGCAATGACGTAGAT |
| TGF-β1 | TCAGACATTCGGGAAGCAGT | ACGCCAGGAATTGTTGCTAT |
| Collagen 4a1 | TTAAAGGACTCCAGGGACCAC | CCCACTGAGCCTGTCACAC |
| MCP-1 | GCCTGCTGTTCACAGTTGC | CAGGTGAGTGGGGCGTTA |
| β-Actin | CAGCTGAGAGGGAAATCGTG | CGTTGCCAATAGTGATGACC |

β-actin, beta-actin; MCP-1, monocyte chemoattractant protein-; TGF-β1, transforming growth factor- β1.

Semiquantitative analysis was performed using a Java based software program Image J (National Institutes of Health, Behesda, MD, USA). The percentage of the stained area relative to the whole area of the field (% area) in each field was determined.

**Frozen section.** 6-µm frozen sections were fixed in acetone and then blocked (Dako, Carpinteria, CA, USA). They were incubated in rat anti-mouse monoclonal antibody F4/80 (dilution 1:100) (MCA497R, ABD Serotec, USA) followed by a secondary HRP tagged goat anti rat antibody (dilution 1:200) (ABD Serotec, USA). Antigen-antibody reactions were visualized with the chromogen diaminobenzidine and counter staining was performed using Mayer's Haematoxylin (Fronine, Taren Point, NSW, AU) followed by Scott's solution (Fronine Taren Point, NSW, AU). Semiquantitative analysis of the area stained was performed as described above.

## SSAO activity

Assays were performed by a commercial service at Tetra-Q ADME, The University of Queensland, Brisbane, Australia on kidney samples from the diabetic mice. The kidney tissue was homogenised in a buffer containing 20mM HEPES, mannitol, sucrose and cOmplete™ Mini p (Roche Diagnostics, Mannheim, Germany) using a tissue disrupter and then centrifuged. The homogenate samples were incubated with clorgyline and pargyline in phosphate buffer; positive control samples contained PXS-4728A. The preparations were then incubated with a $^{14}$C-benzylamine/benzylamine solution (200M, PerkinElmer, Meridin, CT) and the reaction was terminated with citric acid. The oxidized product was extracted with toluene:ethyl acetate with further centrifuging. The lower aqueous layer was frozen with dry ice and the organic layer was transferred to a 20 mL glass scintillation vial. Scintillant (10 mL) was added to each vial and the amount of radioactivity present in the sample determined using a liquid scintillation counter (Beckman LS6500). The study samples were analysed in duplicate.

## Analysis of off target effects

Lung, liver, pancreas, spleen, gonadal, vascular and cardiac tissue were fixed with 4% neutral buffered formalin overnight then 70% ethanol. 5 animals from each group from the chronic diabetic model were used as this represented the longest exposure to the experimental drug. For lung and liver tissue, a ~3mm wedge was resected and embedded into a block in paraffin. Pancreas, spleen, gonadal and vascular tissue were excised a whole specimen without further dissection. The heart was dissected into half lengthways and embedded into a block in paraffin. The block was cut onto multiple glass slides (3µm). Sections were dewaxed in xylene and rehydrated in graded concentrations of ethanol. Sections were stained with Mayer's Haematoxylin (Fronine, Taren Point, NSW, AU) and examined qualitatively using a light microscope as above.

## Statistical analysis

Statistical analysis was done using Graph Prism 6 software. Six-eight animals were used in each group unless otherwise stated. Statistical significance was assessed using unpaired t Test and one way ANOVA. Two way ANOVA test was used to determine ICC for the histological scores. All data are expressed as mean and standard error measurement (SEM). Statistical significance was considered only if the probability value was $<0.05$. Blood glucose profile during the study was measured using repeated measures Anova. Anova with Bonferroni's correction for multiple comparisons was used for all other statistical analysis.

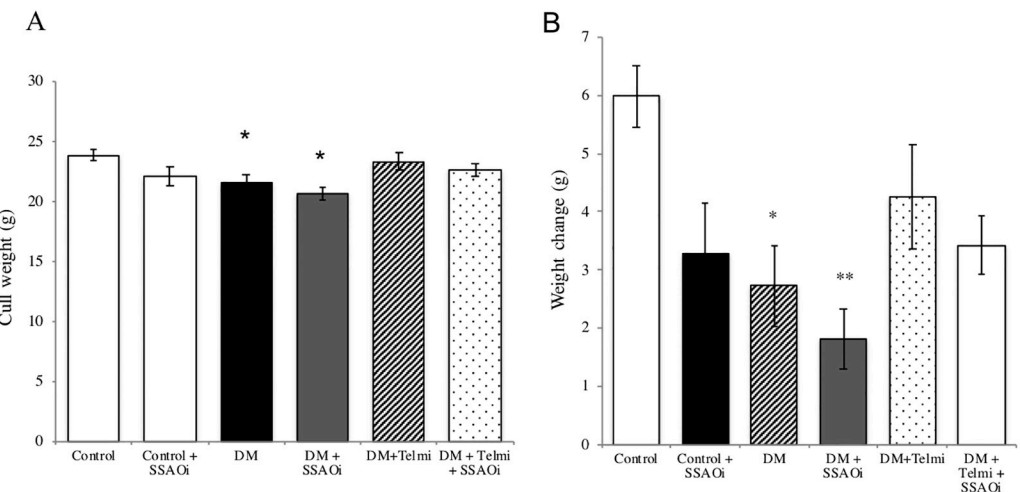

**Fig 1.** SSAO inhibitor induced poor weight gain (Fig A) Cull weight. (Fig B) Weight change from pre-diabetes induction to pre-cull. Diabetic mice had reduced weight gain compared to control. This was similar to diabetic mice treated with SSAO inhibitor. The cull weight was similar across groups. Data expressed as mean ± SEM. (*p<0.05, **p<0.01, vs control.) n = 6–8. DM, Diabetes Mellitus; SSAOi, Semicarbazide Sensitive Amine Oxidase Inhibitor; Telmi, telmisartan.

## Results

### Animal characteristics and physiological parameters

Daily treatment with SSAOi over the 24-week period was well tolerated with no adverse clinical effects noted. Cull weight was reduced in diabetic mice treated with SSAOi compared to untreated diabetic mice (20.7g ± 0.5g vs 21.5g ± 0.72g respectively). Diabetic mice had significantly reduced cull weight when compared to control mice (21.5±0.7g vs 23.9 ± 0.5g respectively p<0.05). Diabetic mice had significantly poorer weight gain compared to control (2.7± 0.7g vs 6.0 ± 0.5g respectively; p<0.05 using unpaired t-tests). This effect was more pronounced in those treated with SSAOi compared to controls (1.8±0.5g vs 6.0 ± 0.5g respectively; p<0.01) (Fig 1).

### Diabetic mice had similar BGL and HbA1c

Diabetic mice displayed significantly elevated blood glucose levels at 20.0 ±0.6mmol/L after induction with streptozotocin. As stated in the methods, animals that had BGL greater than 28mmol/l were maintained with long acting insulin (p<0.001 compared to control). There were no significant differences in blood glucose levels between diabetic mice compared to diabetic mice treated with the SSAO inhibitor (20.7±0.8mmol/L), telmisartan (24.2±1.1 mmol/L), or combination therapy BGL (20.2±1.0mmol/L) (Table 2).

HbA1c levels were significantly increased in all diabetic groups compared to control (p<0.001 vs. Ctrl). There was no significant difference in HbA1c between diabetic groups. Comparisons were made between the diabetic mice receiving a thrice weekly insulin dose of diabetic mice; before and after initiation of SSAOi or telmisartan alone, or in combination. The amount of insulin received in untreated diabetic mice (0.1±0.05IU/wk) compared with the amount of insulin received by mice who received telmisartan therapy (0.3±0.09IU/wk; p<0.01) was statistically significant. Furthermore, SSAOi treated diabetic mice had significantly reduced insulin requirements in comparison with diabetic mice who received

**Table 2. Metabolic and physiological parameters of studied animals.**

| n = 6–8 | CTRL | CTRL+SSAOi | DM | DM+SSAOi | DM+Telmi | DM+Telmi+SSAOi |
|---|---|---|---|---|---|---|
| **Fasting Blood glucose level (mmol/L)** | 9.5±0.44 | 10.2±0.41 | 20.0±0.60* | 20.7±0.77* | 24.2±1.06* | 20.2±1.0* |
| **HbA1c (%)** | 4.3±0.05 | - | 7.0±0.26* | 7.1±0.49* | 7.7±0.40* | 7.9±0.44* |
| **Weekly Insulin requirement (IU)** | - | - | 0.1±0.05 | 0.04±0.0^ | 0.3±0.09# | 0.1±0.04 |
| **Urinary Albumin: Creatinine ratio (μg/mg)** | 113±17 | 104±13 | 362±85 | 126±21## | 132±34## | 172±51# |

*p<0.001 vs control,

#p<0.05

##p<0.01 vs diabetes,

^p<0.05 DM+SSAOi vs DM+Telmi.

CTRL, control; DM, Diabetes Mellitus; SSAOi, Semicarbazide Sensitive Amine Oxidase Inhibitor; Telmi, Telmisartan; HbA1c, glycated haemoglobin A1c.

telmisartan (0.04±0.02IU/wk vs 0.3±0.09IU/wk; p<0.01). There was no difference in insulin requirement between untreated diabetic mice and diabetic mice who received SSAOi.

## Telmisartan and SSAOi reduced urinary albumin to creatinine ratios in diabetic mice

Albuminuria as a marker for glomerular damage was determined in the final week of the experiment. The urinary albumin to creatinine ratio (ACR) was greater in untreated diabetic mice (362±85 μg/mg) compared to the control group (113±17μg/mg; P<0.01 vs. Ctrl). Diabetic mice who received SSAOi (126±21μg/mg) and diabetic mice treated with telmisartan (132±34 μg/mg) had a significant reduction of albuminuria compared to untreated diabetic mice (both P<0.01). There was no statistically significant difference between diabetic mice treated with telmisartan versus SSAOi in urinary albumin to creatinine ratio. Diabetic mice who had treatment with combination therapy (SSAOi+telmisartan) had a significantly lower urinary ACR compared to untreated diabetic mice (172±51 μg/mg, vs 362±85 μg/mg; P<0.05). (Table 2).

## SSAOi improved the glomerulosclerosis index score (GIS) but not the tubulointerstitial damage index (TDI) in diabetic mice

Untreated diabetic mice demonstrated prominent glomerulosclerosis in contrast to control mice (GIS 0.6±0.09 vs 0.06±0.02 respectively; p<0.001). Diabetic mice treated with SSAOi had significantly lower glomerulosclerosis index scores (GIS) compared to untreated diabetic mice (0.42±0.07 vs 0.6±0.09 respectively; p<0.05). Treatment with telmisartan was also associated with improvement in the GIS in diabetic mice (0.40±0.07 vs 0.6±0.09 respectively), ICC>0.70; p<0.05. Despite no synergistic reduction in albuminuria, combination treatment with SSAOi and telmisartan led to the greatest improvement in GIS compared to untreated diabetic mice (0.15±0.03 vs 0.6±0.09 respectively; p<0.001) (Fig 2A).

Diabetic mice developed mild tubular atrophy and dilatation when compared with control mice (0.90±0.1 vs 0.44±0.1;p<0.05). There was no significant reduction in the TDI with SSAOi treatment (0.90±0.2 vs 0.90±0.1) or telmisartan (0.90±0.2 vs 0.90±0.1) in the kidneys of diabetic mice as assessed by Masson's trichrome staining (Fig 2B), ICC>0.45; p = ns.

## SSAOi administration did not alter the renal cortical inflammatory or profibrotic cytokines in diabetic mice

The transcription of the inflammatory cytokines MCP-1 and the ECM component fibronectin were increased in untreated diabetic mice vs control (both p<0.05). The transcription of

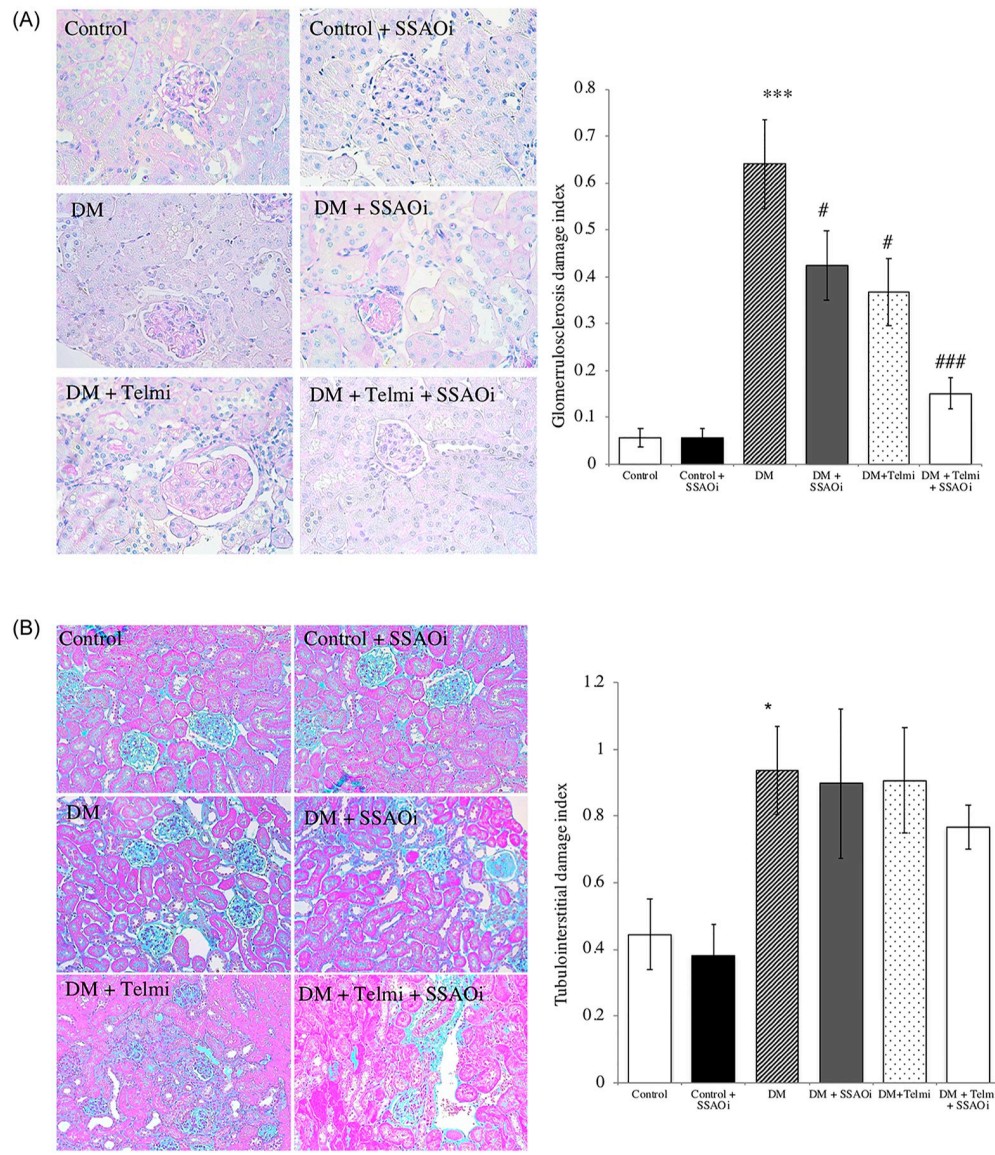

**Fig 2. SSAO inhibition reduced glomerulosclerosis but did not reduce the degree of tubulointerstitial fibrosis in diabetic mice.** (Fig A) Glomerulosclerosis index representative photomicrographs of PAS stained sections of tubulointerstitium and histograms. (Fig B) Tubulointerstitial damage index: diabetic mice demonstrated tubular atrophy and tubulointerstitial fibrosis, Representative photomicrographs of Masson Trichrome sections and histograms tubulointerstitial damage index. Quantification of tubular atrophy in all groups was done by counting the number of atrophic tubules per 400 tubule count. Data presented as mean ± SEM. (***p<0.001 vs ctrl; #p<0.05, ###p<0.001 vs DM). n = 6–8. Bar = 200µm DM, Diabetes Mellitus; SSAOi, Semicarbazide Sensitive Amine Oxidase Inhibitor; Telmi, telmisartan.

MCP-1 and fibronectin was unchanged by SSAOi. Diabetic mice treated with telmisartan had no significant difference in MCP-1 and fibronectin transcription. No change was noted in the transcription of collagen IV between control and diabetic mice (Fig 3A). Telmisartan alone and in combination with SSAOi did not alter the renal cortical transcription of inflammatory or profibrotic cytokines in diabetic mice (Fig 3B and 3C).

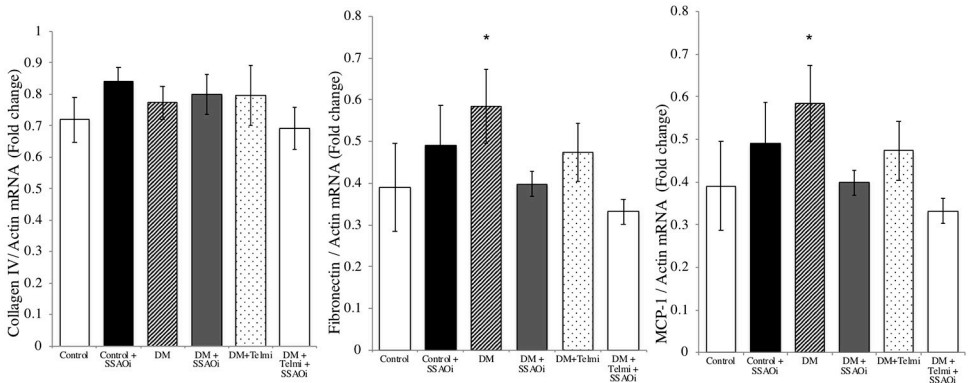

**Fig 3. SSAO inhibition did not improve the renal cortical transcription of inflammatory or profibrotic cytokines in diabetic mice.** Diabetic mice showed increased renal cortical transcription of inflammatory and fibrotic cytokines. Real time PCR results for Renal cortical transcription of (Fig A) Collagen IV, (Fig B) Fibronectin and (Fig C) MCP-1 relative to actin (Data are expressed as mean± SEM). Data presented as mean ± SEM. n = 6–8. DM, Diabetes Mellitus; Telmi, Telmisartan; SSAOi, Semicarbazide Sensitive Amine Oxidase Inhibitor.

## SSAOi administration inhibits cortical transcription fibronectin deposition in diabetic mice but not collagen IV deposition

We assessed fibronectin staining by immunohistochemistry and found the deposition of fibronectin was significantly upregulated in untreated diabetic mice vs control (p<0.05). The percent of cortex staining for fibronectin in untreated diabetic mice was 3.2±0.3% vs 1.23±0.3% in diabetic mice treated with SSAOi (p<0.05). Diabetic mice receiving combination therapy had a more marked reduction in fibronectin deposition when compared to untreated diabetic mice (0.85±0.1% vs 3.2±0.3% area; p<0.001). A significant difference in cortical staining for fibronectin was not found in mice treated with telmisartan (2.4±0.4% staining, p = 0.47 vs control). (Fig 4A). We found a statistically significant increase in collage IV staining in the untreated diabetic mice when compared to control animals (1.0±0.1% vs. 0.54±0.1%; p<0.05). However, there were no differences seen between diabetic mice treated with SSAOi (0.72 ±0.07% or telmisartan (1.0%±0.07%) when compared to untreated diabetic mice (Fig 4B).

## SSAOi suppressed nitrotyrosine expression

Nitrotyrosine is a marker of cell damage, inflammation, and nitric oxide (NO) production. In prior studies we demonstrated that nitrotyrosine is minimally expressed in the kidneys of sham-operated animals but is markedly increased in UUO kidneys [22]. In the current experiment, nitrotyrosine expression, expressed as percentage of the cortex staining positively for nitrotyrosine, was significantly elevated in the diabetic group compared to controls (1.7±0.3% vs 0.3±0.04%; p<0.001). Both telmisartan (0.5±0.05%, p<0.001) and SSAOi (0.6±0.06%, p<0.001) treatment significantly suppressed nitrotyrosine expression, a finding similarly seen with the combination of SSAOi and telmisartan (0.5±0.05% p<0.0001) when compared to untreated diabetic mice (Fig 5A). When we looked at glomerular specific staining, we found that untreated diabetic mice had significantly increased levels compared to control (0.39 ±0.09% vs 0.15±0.05, p<0.05). Diabetic mice treated with SSAOi (0.33±0.08%) and diabetic mice treated with Telmisartan (0.43±0.06%) both had significantly increased levels compared to control (p<0.05 and p<0.01 respectively) (Fig 5B).

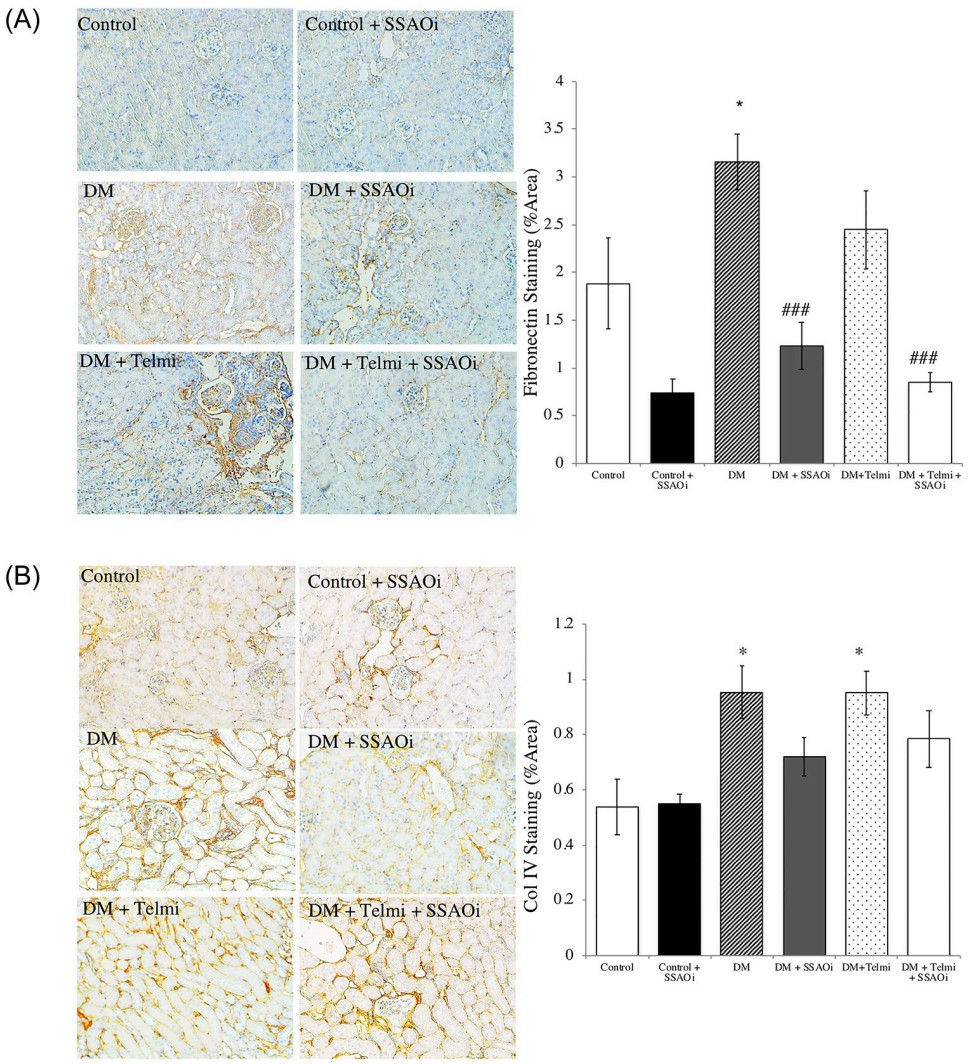

**Fig 4.** Diabetic mice treated with SSAO inhibitor had a reduction in fibronectin but not collagen IV expression (Fig A) Fibronectin staining. Representative photomicrographs of fibronectin staining and histograms summarising expression. Diabetic mice treated with SSAO inhibitor and SSAO/telmisartan combination therapy demonstrated reduced fibronectin staining when compared to untreated diabetic mice. (Fig B) Col IV staining. Representative photomicrographs of collagen IV staining and histograms summarising expression. Untreated diabetic mice and diabetic mice treated with telmisartan exhibited reduced collagen IV staining compared to control mice. Control IgG$_1$ staining was performed to confirm specific staining (not shown). Data presented as mean ± SEM (*p<0.05 vs control; ### p<0.001 vs DM). n = 6–8. Bar = 400μm DM, Diabetes Mellitus; SSAOi, Semicarbazide Sensitive Amine Oxidase Inhibitor; Telmi, telmisartan.

## SSAO inhibition reduces inflammatory cell infiltration

We assessed CD45 staining, a pan-leukocyte marker, and found that the kidneys of diabetic mice had increased expression of CD45 compared to control mice (1.0±0.3% vs 0.26±0.04%; p<0.01). This was significantly reduced when mice were treated with SSAOi (0.43±0.05%, p<0.05). Telmisartan treatment did not lead to a significant difference (0.5±0.07%, p = 0.13), whilst combination treatment resulted in a significant reduction when compared to untreated diabetic mice (0.42±0.04%, p<0.05) (Fig 6A). When we looked at glomerular specific staining,

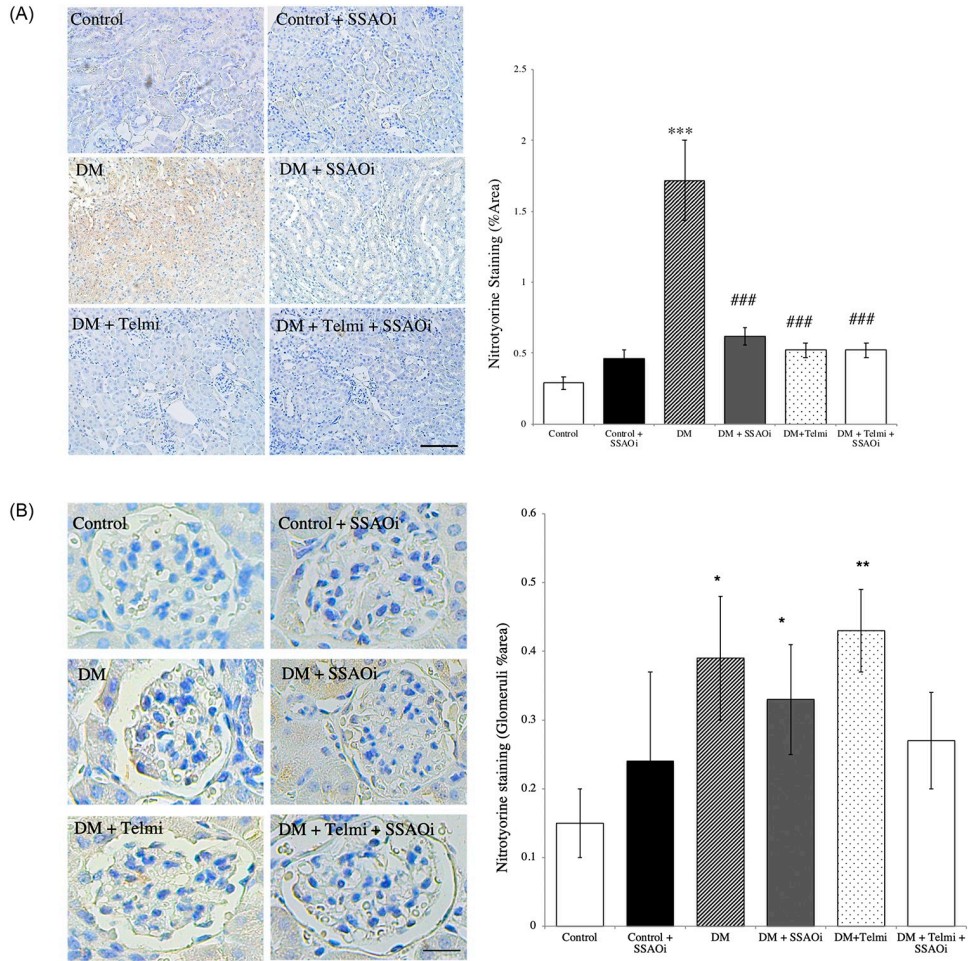

**Fig 5. SSAO inhibition suppressed nitrotyrosine expression.** Representative photographs of nitrotyrosine stained cells and histograms summarising expression. (Fig A) It is minimally expressed in the kidneys of control animals but increased in the kidneys of diabetic mice. Both telmisartan and SSAOi significantly suppressed nitrotyrosine expression, a finding similarly seen with the combination of SSAOi and telmisartan treatment. (Fig B) Nitrotyrosine glomerular specific staining. Control IgG$_1$ staining was performed to confirm specific staining (not shown). Results are expressed as mean ± SE. (***p<0.001 vs control, ###p<0.001 vs DM) n = 6–8. Bar = 400μm DM, Diabetes Mellitus; SSAOi, Semicarbazide Sensitive Amine Oxidase Inhibitor; Telmi, telmisartan.

we found that untreated diabetic mice had significantly increased levels compared to control (0.44±0.08% vs 0.15±0.04, p<0.05) but was not significantly different to other groups of diabetic mice (Fig 6B).

Consistent with reports that macrophages are key players in the pathogenesis of DKD [23], diabetic kidneys in this model showed a trend towards an increase in the macrophage cell surface marker F4/80. F4/80 expression was increased in diabetic mice in comparison to control mice (0.9±0.2% vs 0.42±0.05%; p = 0.06), with no change elicited by treatment with SSAOi, telmisartan or combination therapy (Fig 6C).

## SSAO activity in diabetic kidneys

SSAO activity in untreated diabetic kidneys was no different compared with the control kidneys as determined by the radiolabelled [14C]-benzylamine methods. Diabetic mice treated

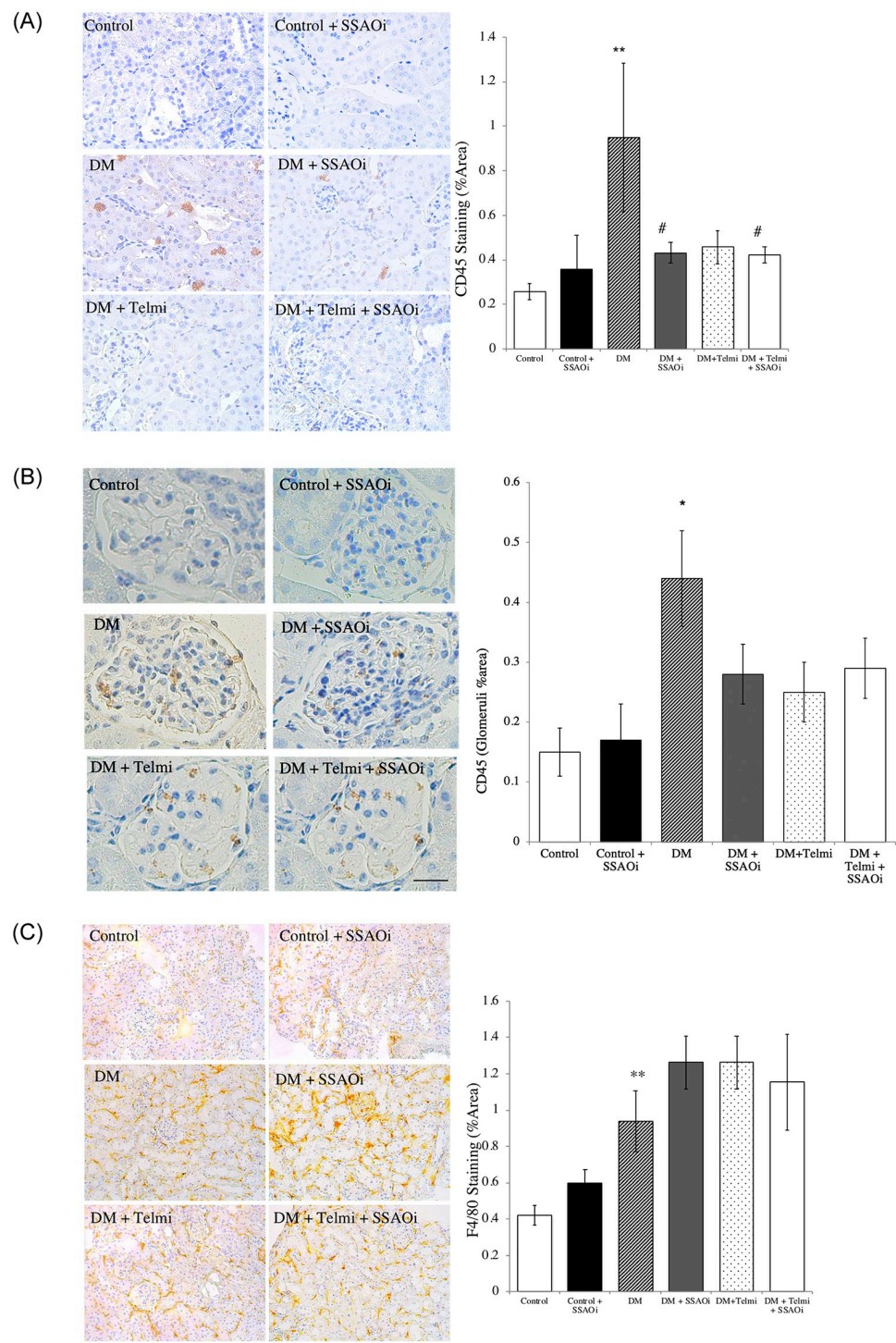

**Fig 6. SSAO inhibition prevents inflammatory cell infiltrates.** (Fig A) CD45 staining, Representative photomicrographs of CD45 positive cells and histograms summarising expression. (Fig B) CD45 glomerular staining, Representative photomicrographs of CD45 positive cells and histograms summarising expression. (Fig C) F4/80 Staining. Representative photomicrographs of immunohistochemistry for tubulointerstitial F4/80 stain for activated macrophages and histograms summarising expression. Control IgG$_1$ staining was performed to confirm specific staining (not shown). Data presented as mean ± SEM with ($^*$p<0.05, $^{**}$p<0.01 vs control, #p<0.05 vs DM) n = 6–8. Bar = 200µm DM, Diabetes Mellitus; SSAOi, Semicarbazide Sensitive Amine Oxidase Inhibitor; Telmi, telmisartan.

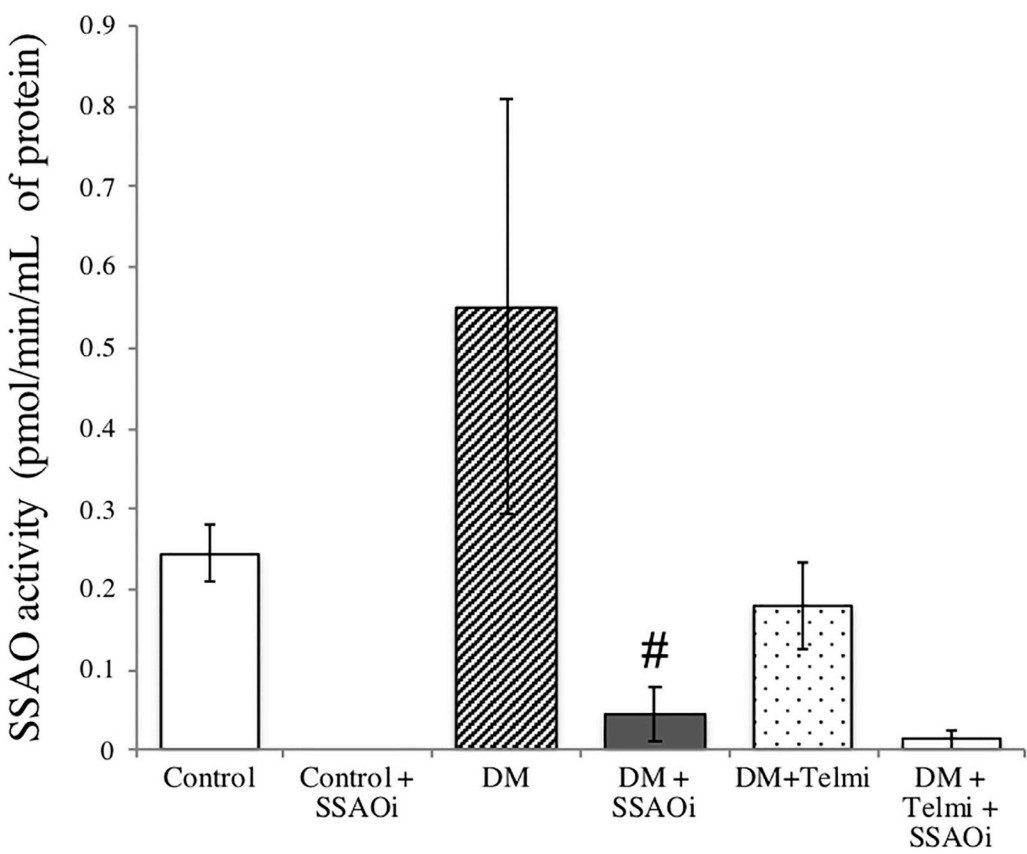

**Fig 7. Radiometric measurement of kidney tissue SSAO activity.** Radiometric measurement of kidney tissue SSAO activity. There was low SSAO activity in the control kidney as determined by the radiolabelled [$^{14}$C]-benzylamine method. Diabetes resulted in increased kidney SSAO activity which is effectively suppressed by administration of an SSAO inhibitor (PXS-4728A) and Telmisartan, as well as combination therapy. Results are represented as mean ± SEM. (#p<0.05 vs DM, ##p<0.001 vs DM) n = 2–10. DM, Diabetes Mellitus; SSAOi, Semicarbazide Sensitive Amine Oxidase Inhibitor; Telmi, telmisartan.

with SSAOi had a significantly lower renal SSAO activity when compared to untreated diabetic mice (0.05 ± 0.03pmol/min/mL of protein vs. 0.55 ± 0.3pmol/min/mL of protein, P <0.05 vs. diabetic control). Mice treated with combination therapy had significantly reduced SSAO activity compared to untreated diabetic mice (0.01 ± 0.01pmol/min/mL of protein vs. 0.55 ± 0.3pmol/min/mL of protein, p = 0<0.001 vs diabetic control when using unpaired t-test). SSAO activity in the diabetic kidneys treated with telmisartan was significantly reduced compared to untreated diabetic kidneys (0.18 ± 0.05pmol/min/mL of protein vs. 0.55 ± 0.3pmol/min/mL of protein, p<0.001 vs diabetic control when using unpaired t-test) (Fig 7).

## SSAO did not cause any off target histological effects in major organs

We found that the organs (lung, pancreas, liver, heart and spleen in control and diabetic animals) of mice treated with the SSAO inhibitor exhibited similar parenchymal architecture. In effect there were no histological changes noted with SSAO inhibition compared with untreated diabetic mice. We also looked at tissues which are known to contain high levels of SSAO activity; there were no architectural changes when comparing untreated diabetic mice to diabetic

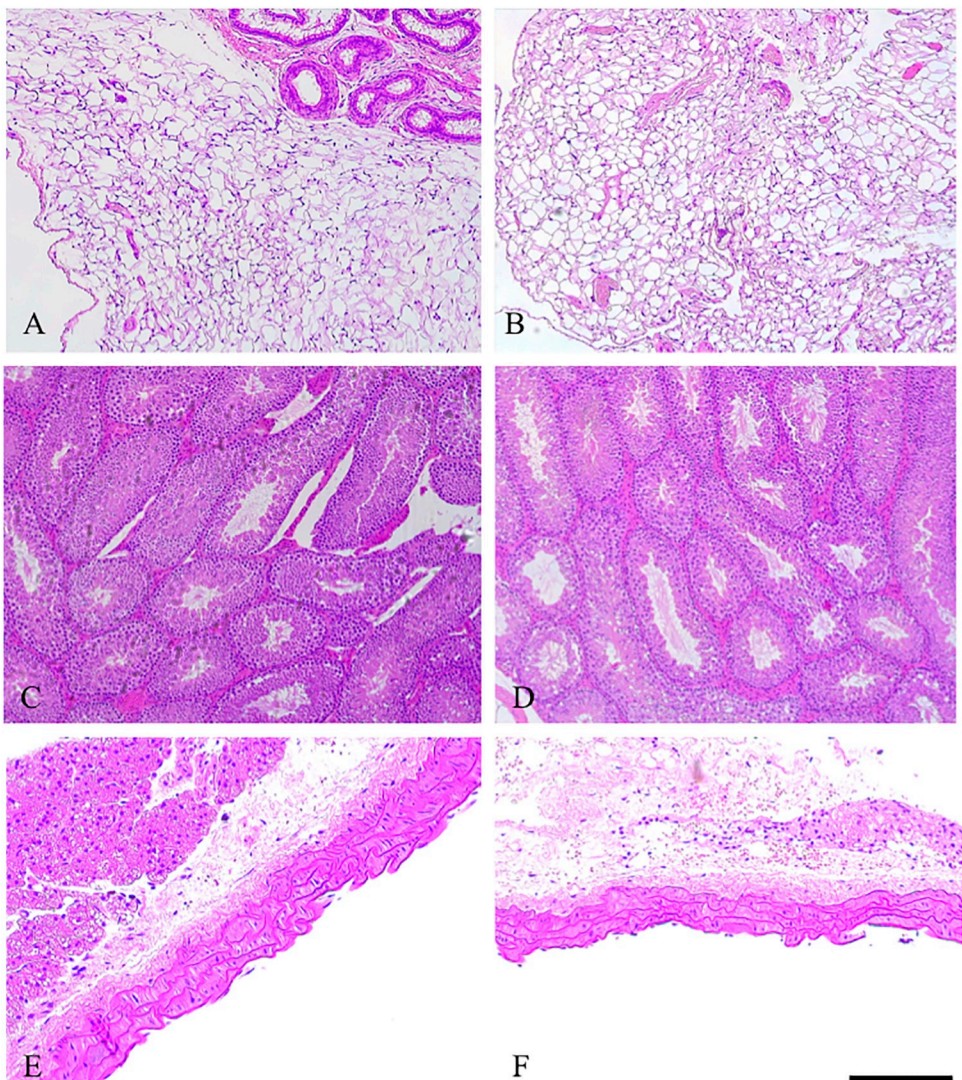

**Fig 8. Representative haematoxylin and eosin staining of other VAP-1 rich organs in control and diabetic animals with and without SSAO inhibitor at 32 weeks (Fig A+B) Bar = 100μm.** (Fig A) DM, (Fig B) DM + SSAOi. (Fig C+D) Bar = 100μm (Fig C) DM (Fig D) DM + SSAOi. (Fig E+F) Bar = 200μm (Fig E) DM (Fig F) DM + SSAOi.

mice receiving SSAO inhibition in fat, gonads or aorta. Fat sections demonstrated heterogenous depots of unilocular adipocytes intercalated with small multi-locular adipocytes as well as interstitial tissue (Fig 8A and 8B). Testicular cross section representing a number of seminiferous tubules composed of germ and Sertoli cells (Fig 8C and 8D). Tunica intima media and adventitia are displayed in the aorta sections (Fig 8E and 8F).

## Discussion

DKD is the most common cause for chronic kidney disease with the end result of end-stage kidney disease. It is also associated with a high frequency of cardiovascular events, including premature mortality, that may occur prior to ESKD occuring [24]. DKD is associated with the

initiation of various pathways including inflammatory pathways that result in the evolution of ESKD. It is known that unresolved inflammation can drive progressive fibrosis [25,26].

We and others have previously demonstrated that diabetic induced eNOS-/- male mice on a C57BL/6 background is a suitable model to study DKD [15–19]. Mice had significantly elevated blood glucose levels after a 5 day induction with streptozotocin, with ketosis prevented by the administration of long acting insulin. Due to protocol driven insulin administration there were no significant differences in blood glucose levels between untreated diabetic mice compared to diabetic mice treated with SSAO inhibitors, telmisartan, or combination therapy. Therefore, there were no significant differences in HbA1c results between the groups. SSAOi therapy (either as monotherapy or in combination therapy) lowered insulin requirements in diabetic mice compared to diabetic mice treated with telmisartan. This can be explained by the SSAO-dependent generation of $H_2O_2$, which stimulates the tyrosine-phosphorylation of insulin receptor substrates (IRS) in the fat cells of rats [27,28]. By inhibiting the dephosphorylation of IRSs, benzylamine can activate the insulin signalling pathway including glucose transport, lipogenesis stimulation and lipolysis inhibition [29]. Furthermore, SSAO substrates catalyse glucose release and movement in fat cells, resulting in hyperglycaemia through GLUT4 translocation [28,30]. Inhibition of SSAO is thought to block the recruitment of GLUT4 to cell surfaces and thereby lower glucose levels. There was no difference when comparing SSAOi therapy with untreated diabetic mice. Insulin requirements were highest in the diabetic group receiving telmisartan, although the mechanism underpinning this observation is unclear. Most studies suggest that telmisartan is an insulin sensitiser through activation of the peroxisome proliferator-activated receptor-γ (PPARγ) [31,32].

Diabetic mice that received SSAO inhibitor had a reduction in urinary ACR compared to untreated diabetic mice. Likewise, diabetic mice that received telmisartan had a significant reduction in urinary ACR compared to untreated diabetic mice. The reduction in urinary ACR in mice treated with the SSAO inhibitor was not significantly different compared to mice treated with telmisartan. These findings are significant as a reduction in albuminuria correlates with improved mortality and a reduction in future ESKD [33]. This corresponds to the reduction in glomerulosclerosis score observed in diabetic mice that received with SSAOi compared to untreated diabetic mice. Combination treatment with SSAOi and telmisartan showed the greatest improvement in glomerulosclerosis index.

Consistent with our findings, humans with elevated soluble SSAO have albuminuria which correlated with the extent of renal impairment [34]. Although speculative, it is possible that SSAOi can lead to further reduction in albuminuria beyond 24 weeks of treatment, consistent with the drug's anti-oxidative stress and anti-inflammatory properties. These findings are similar to a study in the ZSF1 (obese, diabetic and hypertensive) rat DKD model where UD-014, an orally active SSAO inhibitor, demonstrated a significant and dose dependent reduction in albuminuria with a similar potency to losartan. Furthermore, the combination of UD-014 and losartan synergistically reduced albuminuria compared to treatment with either UD-014 or losartan alone [35]. Such results are consistent with those of a randomised, placebo-controlled, phase 2 trial in patients with DKD. After 12 weeks of treatment with ASP8232, a specific soluble SSAOi, patients had a mean 17.7% reduction in urinary ACR. By comparison, patients randomised to placebo had a mean 2.3% rise in urinary ACR versus baseline, giving a significant placebo-adjusted difference of 19.5% between the two groups. The effect of ASP8232 on urinary ACR was greatest at 12 weeks but was still apparent up to 24 weeks after treatment discontinuation. Future longer term studies investigating the effect of ASP8232 on endpoints related to eGFR, albuminuria reduction and hard renal outcomes are warranted to verify whether the drug delays progression of diabetic kidney disease [36].

Abnormal leukocyte accumulation is a recognised driving force in fibrotic diseases. SSAO-deficient mice have demonstrated reduced leukocyte recruitment in inflammatory challenges related to models of autoimmune diabetes and peritonitis, supporting the notion of SSAO in leukocyte transmigration [37]. Studies have shown that SSAO inhibitors impaired leukocyte migration [7,38], suggesting that its locally released, soluble products may promote leukocyte extravasation, acting as direct cross-linkers between immune cells and endothelial cells.

Although the pattern of VAP-1 expression in the kidney was not demonstrated in this study, Tanaka et al. [39] previously demonstrated that VAP-1 is mostly abundant in pericytes. It has also been demonstrated that the kidneys of transgenic mice over-expressing VAP-1 in their endothelium develop glomerulosclerosis [40]. They also found high expression of the receptor for advanced glycosylation end products at the vascular pole of renal corpuscles from mice fed the atherogenic diet. The difference between the Tanaka study and our findings may be related to the difference in models; Tanaka used the kidneys of healthy mice whereas we used kidneys from a diabetic model (endothelial nitric oxide synthase knockout (eNOS$^{-/-}$) mice on a C57BL/6 background). Tanaka et al. demonstrated that VAP-1 inhibition significantly decreases neutrophil aggregation in a rat model of renal ischaemia/reperfusion injury. It remains unclear whether it also inhibits leukocyte aggregation in DKD. However, neutrophil aggregation is not a documented feature of DKD."

Kidneys from diabetic animals exhibited a trend towards an increase in immunohistochemical staining for the macrophage cell surface marker F4/80, consistent with the fact that macrophages are key players in the pathogenesis of diabetic nephropathy. Expression of F4/80 was significantly upregulated in diabetic mice compared to control mice. F4/80 is only one marker of macrophage infiltration and it may be limited in its ability to detect macrophages within mouse glomeruli. We used CD45 staining to detect all leukocytes to provide a representation on overall leukocyte distribution in normal and diseased kidneys. We found that diabetic mice exhibited increased expression of CD45 staining when compared to control mice. This was significantly reduced when mice were treated with SSAOi or combination treatment. We looked at glomerular specific staining to see if we could localise the predominant action of leukocytes. There was an increase in staining for CD45 in diabetic mice compared to control mice. Interestingly, there was no significant difference between control mice and mice treated with SSAOi or combination treatment. This suggests that SSAO inhibition may attenuate leukocyte activity in the glomerulus of diabetic mice. We also looked at glomerular staining for nitrotyrosine and found that there was increase in staining in diabetic mice compared to control mice, as well as in diabetic mice treated with SSAOi and telmisartan. This suggests ongoing production of reactive oxygen species in diabetic mice which are treated with SSAOi and telmisartan. Perhaps other oxidative stress markers might be involved, and this is an area which remains to be assessed.

Because myofibroblast phenotype and function are known to have similarities between different organs, accumulating data now support a scenario that the oxidase activity of SSAO may be an attractive target for therapeutic intervention in fibrosis in multiple tissues, albeit exerting its main effects during the chronic inflammatory stage of the diseases. Collectively, the findings obtained from this study indicate a beneficial effect of SSAOi inhibition in diabetic mice, predominantly protecting the glomerular structure and barrier function. This makes SSAO a promising target in diabetic kidney disease and possibly renal fibrosis of other aetiologies.

SSAO inhibition was associated with a reduction in cortical fibronectin while telmisartan treatment was not. However, despite the reduction in fibronectin protein deposition, no parallel change in fibronectin transcription was observed. Furthermore, the transcription of MCP-1 and collagen IV was increased in diabetic mice and was unchanged by SSAOi. Diabetic mice

treated with telmisartan displayed a trend towards increased MCP-1 mRNA expression, but expression did not significantly diverge from untreated diabetic mice. Our findings that the fibrosis phenotype was milder in mice receiving the SSAO inhibitor support the hypothesis that SSAO enzyme activity plays a key role in fibrogenesis.

Overall tubulointerstitial fibrosis score was not different in any of the treatment groups. Similar to our findings, a previous study by Kosugi et al. detected only partial benefits in the glomerulus but not in the tubulointerstitium of diabetic eNOS$^{-/-}$ mice with telmisartan treatment [41]. A possible explanation for the lack of treatment benefits in the tubulointerstitial compartment may be related to the mouse model used. We used a validated type 1 diabetic eNOS$^{-/-}$ mouse model; thought to target both glomerular and tubular injury. Whilst our group has used C57BL/6 background strains in multiple fibrotic studies–this model may in fact be relatively resistant to the development of diabetic nephropathy. Conversely, mice with a C57BLKS background may have greater sensitivity in developing nephropathy and thus have greater baseline rates of tubulointerstitial fibrosis [14]. We have previously demonstrated in a unilateral ureteric obstruction model that SSAOi ameliorated tubulointerstitial fibrosis with a reduction in Masson's trichrome-stained extracellular matrix with reduced epithelial cell flattening and less tubular damage in comparison to untreated mice with obstructed kidneys [22]. Further studies in chronic disease models associated with more pronounced tubulointerstitial fibrosis may be of value.

The strength of our study is based on the determination of renal parameters in a setting of matched high blood glucose levels among diabetic groups. This study has several limitations. Although the same concentration of telmisartan was used to make a stock solution, there may have been small variability in the amount of telmisartan received due to difficulty in controlling the amount drank by diabetic mice. Furthermore, we did not assess blood pressure. However our group has previously demonstrated no effect of SSAO inhibition and telmisartan on blood pressure in a unilateral ureteric obstruction model [22] as well as in a diabetic eNOS-/- model. We were unable to distinguish the relative contribution of neutrophil aggregation in SSAO inhibition. In addition, the relative contribution of SSAO in the kidney compared to circulating SSAO remains elusive. The exact function of soluble SSAO in the circulation and how it relates to kidney function remains unknown. To overcome these limitations, experiments using organ or tissue specific genetically engineered mice should provide additional information. However, the difficulty in isolating soluble SSAO enzyme activity in our animal experiments compared with human kidneys may attenuate the significance of those results.

## Conclusion

SSAO inhibition in diabetic mice resulted in a significant reduction in glomerulosclerosis and in albuminuria compared to untreated diabetic mice. The combination of RAAS blockade and SSAO inhibition is even more effective in reducing glomerulosclerosis. The effect of SSAOi was less obvious in the tubulointerstitial compartment than in the glomeruli. The results of this study lay the foundation for future clinical studies.

## Acknowledgments

Thank you to the animal facility for their help.

## Author Contributions

**Conceptualization:** Sonia Saad, Muh Geot Wong, Carol Pollock.

**Data curation:** May YW Wong, Sonia Saad, Muh Geot Wong.

**Formal analysis:** May YW Wong, Sonia Saad, Muh Geot Wong, Carol Pollock.

**Funding acquisition:** Sonia Saad, Wolfgang Jarolimek, Heidi Schilter, Carol Pollock.

**Investigation:** May YW Wong, Muh Geot Wong, Heidi Schilter, Carol Pollock.

**Methodology:** May YW Wong, Sonia Saad, Muh Geot Wong, Stefanie Stangenberg, Heidi Schilter, Amgad Zaky, Anthony Gill.

**Project administration:** May YW Wong.

**Resources:** May YW Wong, Sonia Saad, Muh Geot Wong.

**Software:** May YW Wong.

**Supervision:** Sonia Saad, Muh Geot Wong, Carol Pollock.

**Validation:** May YW Wong.

**Writing – original draft:** May YW Wong.

**Writing – review & editing:** May YW Wong, Sonia Saad, Muh Geot Wong, Stefanie Stangenberg, Wolfgang Jarolimek, Heidi Schilter, Amgad Zaky, Anthony Gill, Carol Pollock.

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
