## [Decision Letter · Decision Letter 0]

13 Jan 2020

PONE-D-19-34728

Semicarbazide-sensitive amine oxidase inhibition ameliorates albuminuria and glomerulosclerosis but does not improve tubulointerstitial fibrosis in diabetic nephropathy

PLOS ONE

Dear Dr Wong,

Thank you for submitting your manuscript to PLOS ONE. After careful consideration, we feel that it has merit but does not fully meet PLOS ONE’s publication criteria as it currently stands.

Therefore, we invite you to submit a revised version of the manuscript that addresses the points raised during the review process.

Two expert reviewers have carefully read your manuscript. They have a number of comments that you will ALL need to address in your rebuttal and the revised manuscript.

To enhance the reproducibility of your results, we recommend that if applicable you deposit your laboratory protocols in protocols.io, where a protocol can be assigned its own identifier (DOI) such that it can be cited independently in the future. For instructions see: http://journals.plos.org/plosone/s/submission-guidelines#loc-laboratory-protocols

We look forward to receiving your revised manuscript.

Kind regards,

Jaap A. Joles, DVM, PhD

Academic Editor

PLOS ONE

Journal Requirements:

2. We noticed you have some minor occurrence(s) of overlapping text with the following previous publication(s), which needs to be addressed:

https://doi.org/10.1152/ajprenal.00698.2013

https://doi.org/10.1152/ajprenal.00416.2013

https://doi.org/10.1371/journal.pone.0108994

https://doi.org/10.1371/journal.pone.0141143

https://doi.org/10.1038/srep39863

https://doi.org/10.1038/s41598-018-27462-6

In your revision ensure you cite all your sources (including your own works), and quote or rephrase any duplicated text outside the Methods section. Further consideration is dependent on these concerns being addressed.

3. As part of your revision, please complete and submit a copy of the ARRIVE Guidelines checklist, a document that aims to improve experimental reporting and reproducibility of animal studies for purposes of post-publication data analysis and reproducibility: https://www.nc3rs.org.uk/arrive-guidelines. Please include your completed checklist as a Supporting Information file. Note that if your paper is accepted for publication, this checklist will be published as part of your article.

4. In your Methods section, please give the sources of all reagents used in your study, including telmisartan.

'The authors have declared that no competing interests exist.'

We note that one or more of the authors are employed by a commercial company: Pharmaxis.

Additional Editor Comments (if provided):

Reviewers' comments:

Reviewer's Responses to Questions

**Comments to the Author**

1. Is the manuscript technically sound, and do the data support the conclusions?

Reviewer #1: Yes

Reviewer #2: Yes

2. Has the statistical analysis been performed appropriately and rigorously? 

Reviewer #1: Yes

Reviewer #2: I Don't Know

3. Have the authors made all data underlying the findings in their manuscript fully available?

Reviewer #1: Yes

Reviewer #2: No

4. Is the manuscript presented in an intelligible fashion and written in standard English?

Reviewer #1: Yes

Reviewer #2: Yes

5. Review Comments to the Author

Reviewer #1: This study is significant in establishing a valid DKD model in which the protection of SSAOi against DKD is demonstrated as is the case in the human clinical study. It also proved the amelioration of glomerulosclerosis, which could be inferred from the fact that SSAOi decreases UACR. However, the exploration of the protection mechanism of SSAOi needs some improvement. Although the authors proved that the target of SSAOi-induced protection in this model was the glomeruli, in subsequent experiments they seemed to pursue markers of tubulointerstitial fibrosis and inflammation. It is recommended that the experiments elucidating the protection mechanism of glomeruli are conducted.

Major issues:

1) The localization of SSAO in the DKD model should be elucidated. In our previous study (Tanaka S, et al. Kidney Int. 2017;92: 154-164), we demonstrated that VAP-1 was most abundant in the pericytes in the interstitium of corticomedullary junction in healthy kidneys. In the DKD model employed in this study, the localization might be different from the physiological condition.

2) ROS level and leukocyte infiltration should be evaluated specifically in the glomeruli.

3) The type of leukocytes reduced in the treated diabetic kidneys should be elucidated. In our same previous study, we showed that VAP-1 inhibition significantly decreased neutrophil aggregation. I wonder if this is the case with this model, given the number of macrophages were not significantly changed.

4) The Discussion should be more focused on glomerulosclerosis rather than interstitial fibrosis, given that the tubulointerstitial injury was not significantly different with SSAOi treatment in this model.

Minor issue:

1) Table 2 should include the information of blood pressure.

Reviewer #2: This manuscript describes positive effects of SSAO/VAP-1 inhibitor in diabetic kidney disease in a mouse model. Overall, this is a solid paper. However, its impact in clinical means is compromised as another inhibitor SSAO/VAP-1 inhibitor has shown beneficial effects in clinical trials.

Major:

All figure legends need information about the number of animals used.

Many Figures are not sharp and of poor quality. Please, replace. Figure 8 especially does not give any information, as the histology is really bad in several panels.

Figures need scale bars. Mentioning of the original magnification becomes irrelevant.

There are several inconsistencies, which need to be fixed:

1. Line 72: VAP-1/SSAO is not present on leukocytes

2. Line 76: VAP-1 is Vascular Adhesion Protein-1, not Vascular-antigen peptide

3. Line 282: BSL should be BGL

4. Line 311: The authors write: the urinary albumin to creatinine ratio (ACR) was greater in untreated diabetic mice (362±85 μg/mg) compared to the control group (113±17μg/mg; P<0.01 vs. Ctrl). How and why there is albumin in the urine of healthy mice??????

5. Line 365: SSAOi administration inhibits cortical transcription fibronectin

deposition in diabetic mice but not collagen IV deposition. Remove transcription from the title, because you just analyse fibronectin staining

6. Line 382: Representative photomicrographs of

fibronectin staining cells and histograms summarising expression. Fibronectin is an extracellular matrix protein, remove cells.

7. Line 404: Nitrotyrosine is a marker of cell damage, inflammation, and nitric oxide (NO) production. This type of sentence is already in the text and does not belong to a figure legend.

8. Line 432: Macrophage cells are just macrophages

9. Line 446: The authors write: SSAO activity in the diabetic kidneys treated with telmisartan was no different compared to untreated diabetic kidneys (Fig 7). However, in the figure legend they write: Diabetes resulted in increased kidney SSAO activity which is effectively suppressed by administration of an SSAO inhibitor (PXS-4782A) and Telmisartan, as well as combination therapy. Please, correct.

10. Line 503: The can should be this can

11. Line 508 onwards: Benzylamine is an artificial substrate for SSAO that does not occur naturally. It is used in enzymatic activity measurement. Thus, it cannot contribute in this scenario, if not given exogenously. Please, modify.

12. Line 559 onwards: Delete Nitrotyrosine paragraph – it is simply result no discussion.

13. References: please check the references – there are several inconsistences

6. PLOS authors have the option to publish the peer review history of their article (what does this mean?). If published, this will include your full peer review and any attached files.

Reviewer #1: No

Reviewer #2: No

---

## [Author Response · Author response to Decision Letter 0]

6 May 2020

6th April 2020

Jaap A. Joles, DVM, PhD

Academic Editor

PLOS ONE

Dear Dr Joles and reviewers,

Re: PONE-D-19-34728 Semicarbazide-sensitive amine oxidase (SSAO) inhibition ameliorates glomerulosclerosis but does not improve tubulointerstitial fibrosis in diabetic nephropathy 

Thank you for the opportunity to make changes to our manuscript. We hope it meets your requirements.

Kind regards,

A/Prof. Sonia Saad: 

on behalf of: May YW Wong, Muh Geot Wong, Stefanie Stangenberg, Wolfgang Jarolimek, Heidi Campos Schilter, Amgad Zaky, Anthony J. Gill, Carol Pollock

---

## [Decision Letter · Decision Letter 1]

20 May 2020

PONE-D-19-34728R1

Semicarbazide-sensitive amine oxidase inhibition ameliorates albuminuria and glomerulosclerosis but does not improve tubulointerstitial fibrosis in diabetic nephropathy

PLOS ONE

Dear Dr Wong,

Thank you for submitting your manuscript to PLOS ONE. After careful consideration, we feel that it has merit but does not fully meet PLOS ONE’s publication criteria as it currently stands. Therefore, we invite you to submit a revised version of the manuscript that addresses the points raised during the review process.

Both reviewers are satisfied with your revision and accept the content.  However, I am granting minor revison because once the manuscript is accepted you have no opportunity to change anything. Note that PLoS One also no longer sends out proofs! Therefore you must first check all the references. Reviewer #2 remarks that some are incomplete or the cited papers have appeared in print and are no longer an ePub in press.

We would appreciate receiving your revised manuscript by Jul 04 2020 11:59PM. To enhance the reproducibility of your results, we recommend that if applicable you deposit your laboratory protocols in protocols.io, where a protocol can be assigned its own identifier (DOI) such that it can be cited independently in the future. For instructions see: http://journals.plos.org/plosone/s/submission-guidelines#loc-laboratory-protocols

We look forward to receiving your revised manuscript.

Kind regards,

Jaap A. Joles, DVM, PhD

Academic Editor

PLOS ONE

Reviewers' comments:

Reviewer's Responses to Questions

**Comments to the Author**

1. If the authors have adequately addressed your comments raised in a previous round of review and you feel that this manuscript is now acceptable for publication, you may indicate that here to bypass the “Comments to the Author” section, enter your conflict of interest statement in the “Confidential to Editor” section, and submit your "Accept" recommendation.

Reviewer #1: All comments have been addressed

Reviewer #2: All comments have been addressed

2. Is the manuscript technically sound, and do the data support the conclusions?

Reviewer #1: Yes

Reviewer #2: Yes

3. Has the statistical analysis been performed appropriately and rigorously? 

Reviewer #1: I Don't Know

Reviewer #2: Yes

4. Have the authors made all data underlying the findings in their manuscript fully available?

Reviewer #1: Yes

Reviewer #2: Yes

5. Is the manuscript presented in an intelligible fashion and written in standard English?

Reviewer #1: Yes

Reviewer #2: Yes

6. Review Comments to the Author

Reviewer #1: (No Response)

Reviewer #2: Please, check once more the references: some are missing page numbers and some newer references have now come out as prints.

7. PLOS authors have the option to publish the peer review history of their article (what does this mean?). If published, this will include your full peer review and any attached files.

Reviewer #1: No

Reviewer #2: No

---

## [Author Response · Author response to Decision Letter 1]

29 May 2020

We have verified the dates and page numbers for references

---

## [Editor Report · Decision Letter 2]

1 Jun 2020

Semicarbazide-sensitive amine oxidase inhibition ameliorates albuminuria and glomerulosclerosis but does not improve tubulointerstitial fibrosis in diabetic nephropathy

PONE-D-19-34728R2

Dear Dr. Wong,

We are pleased to inform you that your manuscript has been judged scientifically suitable for publication and will be formally accepted for publication once it complies with all outstanding technical requirements.

With kind regards,

Jaap A. Joles, DVM, PhD

Academic Editor

PLOS ONE
---

## [Editor Report · Acceptance letter]

4 Jun 2020

PONE-D-19-34728R2 

Semicarbazide-sensitive amine oxidase inhibition ameliorates albuminuria and glomerulosclerosis but does not improve tubulointerstitial fibrosis in diabetic nephropathy 

Dear Dr. Wong:

I'm pleased to inform you that your manuscript has been deemed suitable for publication in PLOS ONE. Congratulations! Your manuscript is now with our production department. 

Kind regards, 

on behalf of

Dr. Jaap A. Joles 

Academic Editor

PLOS ONE